# Mortality Trends in Alzheimer’s Disease in Mississippi, 2011–2021

**DOI:** 10.3390/diseases11040179

**Published:** 2023-12-11

**Authors:** Elizabeth A. K. Jones, Brenda Jenkins, Clifton Addison

**Affiliations:** 1School of Public Health, College of Health Sciences, Jackson State University, Jackson, MS 39213, USA; brenda.w.campbell@jsums.edu (B.J.); clifton.addison@jsums.edu (C.A.); 2Jackson Heart Study Graduate Training & Education Center, Jackson State University, Jackson, MS 39213, USA

**Keywords:** Alzheimer’s, Alzheimer’s disease, mortality, death rates, adults, seniors, dementia

## Abstract

Alzheimer’s disease is the sixth most common cause of death in the United States (U.S.), with one in three adults 65 years of age and older dying of the disease each year. Deaths from Alzheimer’s have more than doubled between 2000 and 2019, killing more adults than both breast cancer and prostate cancer. In 2021, Alzheimer’s disease resulted in 36 deaths per 100,000 in the U.S. In Mississippi, deaths from Alzheimer’s have almost doubled between 2011 and 2021, resulting in 52.9 deaths per 100,000. Women have a higher mortality rate from Alzheimer’s than men. Alzheimer’s is a progressive disease that develops through seven stages. There are effective strategies to prevent the onset of Alzheimer’s. Methods: This paper reviews the risk factors, mortality trends, etiology, and prognosis of Alzheimer’s in Mississippi with a focus on prevention. Results: The southern diet with foods high in sugar and sodium, along with sedentary and poor lifestyle choices, increases mortality risk from Alzheimer’s disease for women in Mississippi, specifically due to women over 65 having higher rates of obesity and hypertension. Conclusion: Understanding the epidemiology and risk factors of Alzheimer’s in Mississippi will help inform communities, policies, and programs to prevent disease occurrence.

## 1. Introduction

Alzheimer’s disease is the most common type of dementia [1]. Alzheimer’s is characterized as a progressive form of memory loss that begins as mild memory loss and advances to loss of verbal communication skills and an inability to respond within an environment [1]. Alzheimer’s affects the brain, specifically areas that are related to thought, memory, and language [1]. The probable causes of Alzheimer’s disease include (1) age-related changes in the brain, (2) genetic factors, (3) environmental factors, and (4) lifestyle factors [2]. Numerous research findings have suggested that lifestyle factors play a major role in the onset of Alzheimer’s disease [3,4,5,6].

According to the Centers for Disease Control and Prevention, 5.8 million people living in the United States were living with Alzheimer’s disease, with 134,242 deaths due to Alzheimer’s disease in 2020 [1]. In the United States, Alzheimer’s disease is the sixth leading cause of death and the fifth leading cause of death among people 65 and older living in the United States [7]. Approximately one in three seniors dies of Alzheimer’s in the United States [8].

In Mississippi, it is estimated that the number of people living with Alzheimer’s disease aged 65 and older will increase by 18.2% by 2025 to 65,000 [9]. However, Alzheimer’s cases could be prevented by implementing healthier lifestyle choices, such as exercising at least 3 h per week, eating a Mediterranean-style diet, reducing “empty calorie intake” (i.e., bread, pasta, rice, and sweets), minimizing alcohol intake, getting adequate sleep, engaging in stress reduction techniques, protecting oneself from head injuries, managing hearing loss, staying social, learning new things, and visiting your primary care provider regularly [10]. The incidence of and mortality from Alzheimer’s disease is highest in white women aged 65 years and older [11]. In Mississippi, 56.3% of mortality from Alzheimer’s disease occurs in white women 65 years of age and older. Furthermore, Mississippi has a significantly higher mortality rate than the U.S. average. In Mississippi, 52.9 per 100,000 deaths resulted due to Alzheimer’s, while the U.S. average was 36.0 deaths per 100,000 in 2021 [12].

The common symptoms are issues with memory, thinking, and reasoning; making judgments and decisions; planning and performing familiar tasks; and changes in personality and behavior [13]. The late-stage or advanced stage of Alzheimer’s results in required and complete assistance with daily care, loss of awareness of recent experiences and surroundings, changes in mobility, including walking, sitting, and swallowing, difficulty in communicating, and susceptibility to infections such as pneumonia [9]. Alzheimer’s is not detectable by one single test. It is detected using several diagnostic tools, including neurological exams, cognitive and functional assessments, brain imaging (i.e., MRI, CT, PET), and cerebrospinal fluid or blood tests, to make an accurate diagnosis [9]. Adults with the condition will have abnormal results for the various diagnostic tools. Smoking, diets high in sugar and sodium, physical inactivity, and excessive drinking influence the development and progression of Alzheimer’s disease. Despite the long-lasting impact of Alzheimer’s disease, no studies have evaluated mortality trends within the United States or within Mississippi. Therefore, there is a need for an in-depth analysis of the epidemiology of Alzheimer’s disease, Mississippi mortality trends, and the evaluation of risk factors to better understand the role of epigenetics in disease formation and progression and to inform prevention efforts.

## 2. Epidemiology of Alzheimer’s Disease and Mortality Trends

Between 2011 and 2021, Mississippi reported a total of 15,358 Alzheimer’s deaths. The source population from which mortality rates between 2011 and 2021 were derived was obtained from the Mississippi statistically automated health resource system. Most cases were reported among females (70.9%), whites (79.5%), and adults 65 years and older (98.9%) (Figure 1 and Figure 2). The overall age-adjusted mortality rate increased from 33.5 deaths per 100,000 in 2011 to 52.9 deaths per 100,000 in 2021 (a 36.7% increase). In the United States, the age-adjusted mortality rate also increased between 2011 and 2021 (34% increase). However, Mississippi had a higher increase in age-adjusted mortality (36.7% increase) between 2011 and 2021 than the United States (34% increase). In 2011, the age-adjusted mortality rate in the United States was 24.7 deaths per 100,000, which is lower than the age-adjusted mortality rate in Mississippi in 2011 (33.5 deaths per 100,000) [14]. In 2021, the age-adjusted mortality rate in the United States was also lower than the age-adjusted mortality rate in Mississippi, with Alzheimer’s deaths causing only 36 deaths per 100,000 in the United States and 52.9 deaths per 100,000 in Mississippi [14].

According to the Centers for Disease Control and Prevention, southern states had higher age-adjusted mortality rates associated with Alzheimer’s disease than any other states within the United States in 2021. These southern states include Texas (41.9 deaths per 100,00), Arkansas (43.2 deaths per 100,000), Louisiana (42.9 deaths per 100,000), Mississippi (52.9 deaths per 100,000), Alabama (46.8 deaths per 100,000), Georgia (44.5 deaths per 100,000), and South Carolina (40.9 deaths per 100,000) [15]. Of the seven states with the highest age-adjusted Alzheimer’s mortality rates, Mississippi had the highest of the seven states. Explanations for these regional disparities in mortality rates include non-modifiable risk factors, such as lack of a balanced diet, consuming foods high in sugar, consuming foods high in sodium, smoking, excessive drinking, and poor management of medical conditions, such as obesity in mid-life, diabetes, hypertension, stroke, heart problems, high cholesterol, and depression. All of these risk factors are most common within southern states, especially in Mississippi, which increases the risk of Alzheimer’s disease and mortality (Figure 3) [16,17,18,19].

### 2.1. Alzheimer’s Mortality by Gender

Between 2011 and 2021, among males, age-adjusted Alzheimer’s mortality rates increased by 10.8% (30.7 deaths per 100,000 to 34.4 deaths per 100,000), with an average annual increase of 4.61% (AAPC, 4.61%, 95% CI, 1.1% to 9.5%). In females, there was a 43.9% increase in Alzheimer’s mortality rates (34.9 deaths per 100,000 to 62.2 deaths per 100,000), with an average annual increase of 6.78% (AAPC, 6.78%, 95% CI, 4.50% to 9.94%).

The trends in males and females consisted of 1 segment (2011–2021): a significant APC of 4.61% (95% CI, 1.1% to 9.5%) in males and a significant APC of 6.78% in females (95% CI, 4.50% to 9.94%). See Table 1 and Figure 4.

In Mississippi, both males and females had increases in both age-adjusted Alzheimer’s mortality rates and mortality trends. However, the magnitude of the increases varied by gender. Females had higher increases in age-adjusted mortality and mortality trends than males. Explanations for these gender disparities in mortality rates include higher rates of complications with diabetes and hypertension and a greater risk of small vessel heart disease, coronary heart disease, and depression in females [20], all of which increase the risk of Alzheimer’s disease and mortality.

### 2.2. Alzheimer’s Mortality by Race

Between 2011 and 2021, among blacks, age-adjusted Alzheimer’s mortality rates increased by 42.0% (27.3 deaths per 100,000 to 47.1 deaths per 100,000), with an average annual increase of 7.32% (AAPC, 7.32%, 95% CI, 4.89% to 10.97%). In whites, there was a 35.8% increase in Alzheimer’s mortality rates (35.9 deaths per 100,000 to 55.9 deaths per 100,000), with an average annual increase of 5.90% (AAPC, 5.90%, 95% CI, 2.80% to 10.07%). In other races, there was a 32.8% increase in Alzheimer’s mortality rates (7.8 deaths per 100,000 to 11.6 deaths per 100,000), with an average annual increase of 4.19% (AAPC, 4.19%, 95% CI, −5.5% to 20.2%).

The trends in blacks consisted of 1 segment (2011–2021) with a significant APC of 7.32% (95% CI, 4.89% to 10.97%) during the 1st segment. The trends in whites consisted of two segments: a significant APC of 5.90% (95% CI, 2.80% to 10.07%) during the 1st segment (2011–2021) and a significant APC of 6.0% (95% CI, 0.1% to 9.6%) during the 2nd segment (2019–2021). The trends in other races consisted of 2 segments: a significant APC of 35.0% (95% CI, 7.1% to 236.4%) during the 1st segment (2011–2014), and a significant APC of −6.8% (95% CI, −27.7% to −0.9%) during the 2nd segment (2014–2021). See Table 1 and Figure 5.

These findings indicate increases in Alzheimer’s age-adjusted mortality rates in all racial groups between 2011 and 2021, despite the magnitude of the increases for various racial groups differing. In Mississippi, blacks (42.0%) had higher increases in Alzheimer’s mortality rates than whites (35.8%) and other races (32.8%). Blacks also have a higher mortality trend rate than whites and other races in Mississippi. Explanations for these differences in the magnitude of increases in mortality rates include the prevalence of health conditions among blacks, such as heart disease, discrimination within the healthcare system, and issues navigating the healthcare system [21].

### 2.3. Alzheimer’s Mortality by Age Groups

Between 2011 and 2021, among age groups 0–44 years, there were no reported deaths associated with Alzheimer’s disease. Among adults aged 45–64, age-adjusted Alzheimer’s mortality rates increased by 40% (0.3 deaths per 100,000 to 0.5 deaths per 100,000), with an average annual increase of 5.99% (AAPC, 5.99%, 95% CI, 2.2% to 11.2%). In adults 65 years and older, there was an 81.8% increase in Alzheimer’s mortality rate (33.2 deaths per 100,000 to 182.3 deaths per 100,000), with an average annual increase of 6.1% (AAPC, 6.1%, 95% CI, 2.7% to 10.7%).

The trends in adults 45–64 years consisted of one segment (2012–2021); a significant APC of 5.99% (95% CI, 2.2% to 11.2%) occurred during the 1st segment (2012–2021). The trends in adults 65 years and older also consisted of one segment, with a significant APC of 6.1% (95% CI, 2.7% to 10.7%) during the 1st segment (2011–2021). See Table 1.

These findings indicate an increasing trend in Alzheimer’s deaths in adults 45 years of age and older between 2011 and 2021. However, adults 65 years of age and older have a significantly higher mortality rate trend for Alzheimer’s disease than other age groups. Explanations for age disparities in mortality rates include natural changes in the brain, such as brain shrinking in certain areas of the brain, less effective communication between neurons, less blood flow in the brain, and increases in inflammation [22].

## 3. Etiology

The main triggers that cause Alzheimer’s disease are complex [23]. However, the underlying element is the buildup of two substances within the brain called amyloid and tau [23]. When there are issues in the brain, these two substances merge and form tiny structures called plaques and tangles [23]. The plaques and tangles make it difficult for the brain to function properly [23]. As time passes, the disease causes certain parts of the brain to become smaller [23]. It also reduces the number of important chemicals needed to transport messages in the brain [23]. These issues lead to brain damage and memory and thinking problems [23]. Environmental factors, such as exposure to aluminum, exposure to too little or too much zinc, toxins from foodborne illnesses, and viruses, may all lead to the development of Alzheimer’s disease [24].

Other risk factors for Alzheimer’s include lifestyle factors such as diet (high in sodium and sugar), physical inactivity, excessive drinking, smoking, obesity, unmanaged hearing loss, and inadequate sleep. Notably, ultra-processed, high-sodium foods have been identified as a major risk factor for Alzheimer’s disease in adults [25,26,27,28,29,30]. These foods include potato chips, salty snacks, deep-fried and packaged meats, and bottled condiments [25]. A large cohort study conducted for 10 years in the United Kingdom among 72,803 adults ages 55 and older found that nearly 43% more people in the highest group of ultra-processed high sodium food intake developed dementia than in the lowest group [25]. The study also found that for every 10% increase in ultra-processed high-sodium food consumption, the risk of Alzheimer’s rose 14% [25].

## 4. Risk Factors

### 4.1. Non-Modifiable Risk Factors

Non-modifiable risk factors are unchangeable risk factors associated with a condition [31]. There are three non-modifiable risk factors for Alzheimer’s disease. These risk factors are age, sex, and genetics [31]. Age is the most significant risk factor because adults 65 years of age and older have a greater risk of being diagnosed with Alzheimer’s disease than any other age group [31]. Sex is also a risk factor because women ages 65 and older are two times more likely to be diagnosed with Alzheimer’s disease than men ages 65 and older [31]. Genetics is also a risk factor because individuals with familial genes, inherited genes passed on from a parent to a child, or risk genes that increase the susceptibility to developing Alzheimer’s disease increase the risk of diagnosis [31].

### 4.2. Modifiable Risk Factors

Modifiable risk factors are changeable factors [31]. There are two major categories of modifiable risk factors for Alzheimer’s disease. The two categories are lifestyle factors and health conditions [31]. A healthy lifestyle, such as not excessively drinking, smoking, and eating a balanced diet that is low in sugar and salt, can reduce the risk of Alzheimer’s [31]. Findings suggest that adults ages 40–65 who live a healthy lifestyle are less likely to be diagnosed with Alzheimer’s disease [31].

Certain health conditions are also considered a risk factor for Alzheimer’s disease [31]. Diabetes, stroke, heart problems, hypertension, high cholesterol, and obesity in mid-life, age-related hearing loss, and depression increase the likelihood of being diagnosed with Alzheimer’s disease [31]. However, actively managing these conditions and seeing a health care provider regularly can reduce the risk of being diagnosed with Alzheimer’s disease [31].

## 5. Prevention

Non-modifiable risk factors, such as age, sex, and genetics, cannot be controlled. However, changes to lifestyle habits can reduce the risk of developing Alzheimer’s disease. Managing dietary intake of sugars, sodium, and other unhealthy foods/habits, sleep times, and weight can significantly reduce the risk of developing the disease. The implementation of healthier lifestyle choices can especially improve outcomes when implementing healthier habits at ages 40–65 years and onward [31].

While there is no cure for Alzheimer’s disease, there are numerous treatments, such as galantamine, rivastigmine, and donepezil, which are used to reduce cognitive and behavioral symptoms [32]. There is also a new treatment recently approved by the U.S. Food and Drug Administration (FDA), which has been shown to reduce cognitive and functional decline in individuals living with early Alzheimer’s [33]. The progression of the disease can also be impeded by implementing (1) lifestyle changes, such as maintaining a healthy diet, (2) exercising, (3) maintaining good heart and vascular health by not smoking, controlling your cholesterol, maintaining normal blood glucose levels, maintaining a healthy weight, relaxing and avoiding stress, (4) socializing, and (5) engaging in mentally stimulating activities, such as playing board games, solving puzzles, learning a new language, or playing an instrument [34].

## 6. Conclusions

While age is the most significant risk factor for Alzheimer’s disease, lifestyle choices and health conditions also play a major role in Alzheimer’s mortality rates. In Mississippi, whites have higher rates of hypertension than any other race and higher rates of Alzheimer’s disease, which highlights the role a healthy diet that is low in sugar and sodium plays in Alzheimer’s diagnosis and mortality (Figure 6). Based on the study’s findings, hypertension rates stratified by racial groups follow the established pattern for Alzheimer’s mortality risk, with whites having the highest rates of hypertension and Alzheimer’s mortality, especially white women, followed by blacks and other races.

However, morality trends among racial groups demonstrated that the racial risk factor may be shifting toward blacks having higher mortality rates than whites. As the APC for blacks was 1.37% higher than whites, this depicts a trend toward higher mortality rates in blacks compared to whites. This finding is fairly uncommon, as only a few studies have identified a possible change in racial demographics for the risk of Alzheimer’s disease.

A good understanding of Alzheimer’s mortality risk, associations, and risk factors, as well as a detailed analysis of the condition, will be useful to public health practitioners in decision-making and policy development. Understanding these concepts will also help inform prevention and intervention programs and strategies. The best recommendations to reduce the risk of Alzheimer’s disease are to: (1) practice good lifestyle choices; (2) manage health conditions; (3) get enough sleep; and (4) know your risks.

## Figures and Tables

**Figure 1 diseases-11-00179-f001:**
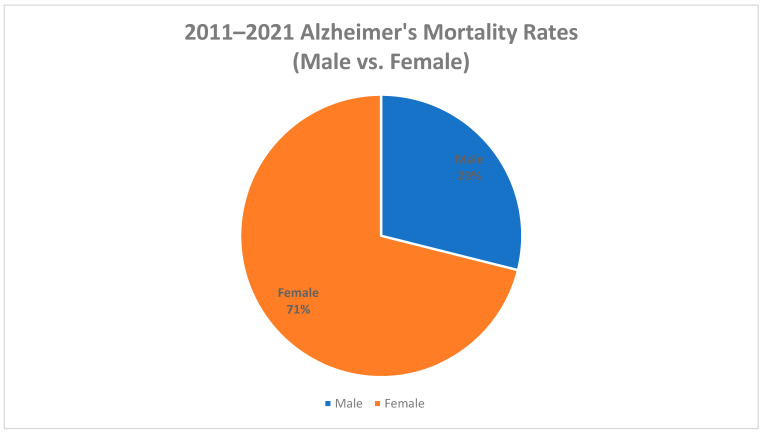
Percentage of Alzheimer’s deaths by gender, Mississippi, 2011–2021.

**Figure 2 diseases-11-00179-f002:**
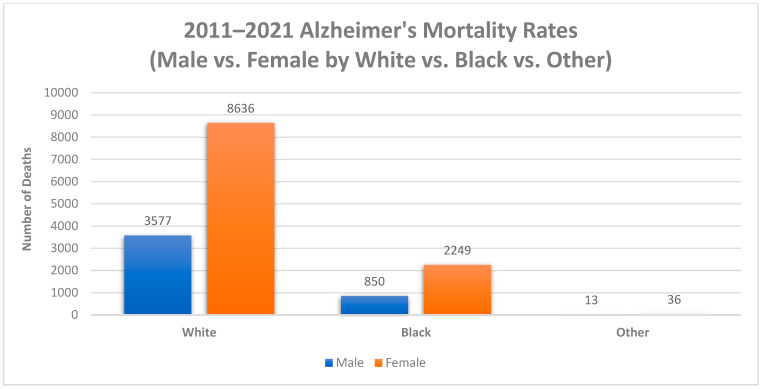
Number of Alzheimer’s deaths by Sex by Race, Mississippi, 2011–2021.

**Figure 3 diseases-11-00179-f003:**
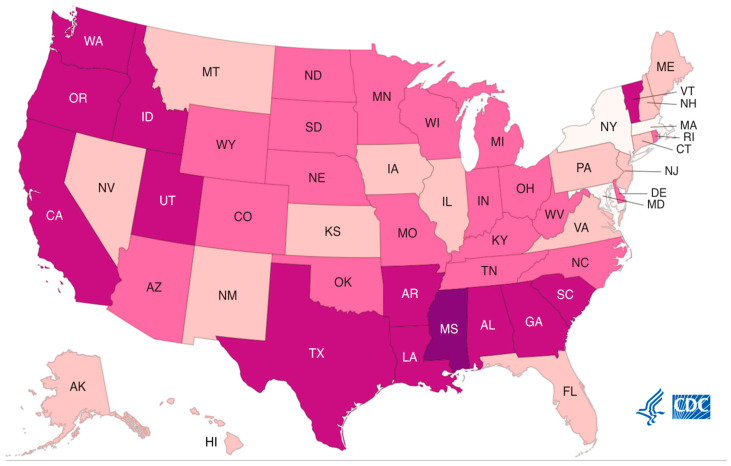
Alzheimer’s mortality, USA, 2021. (As cases per state increase shades darken).

**Figure 4 diseases-11-00179-f004:**
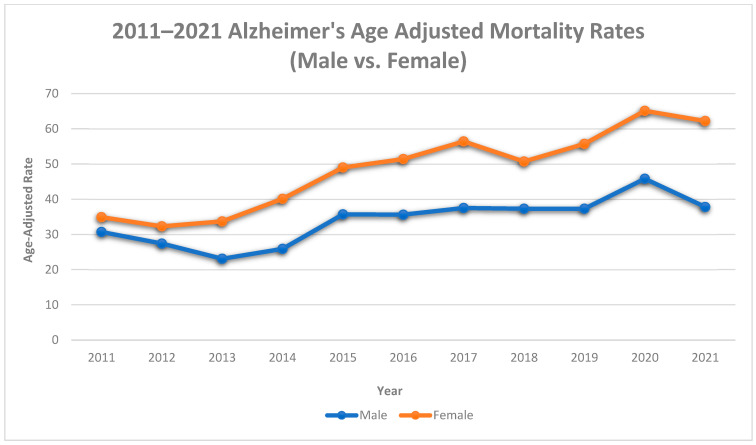
Alzheimer’s age-adjusted mortality rates by gender, Mississippi, 2011–2021.

**Figure 5 diseases-11-00179-f005:**
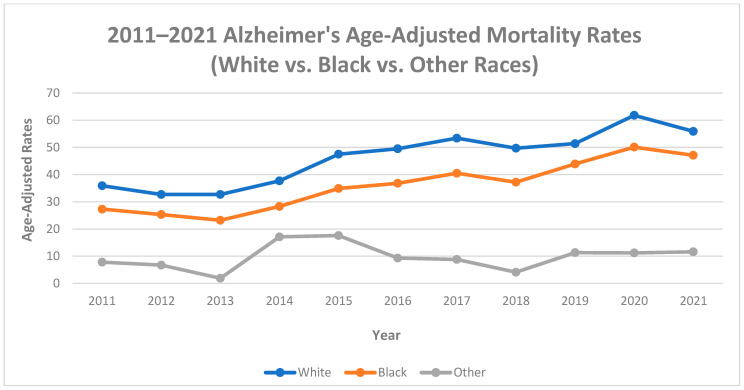
Alzheimer’s age-adjusted mortality rates by race, Mississippi, 2011–2021.

**Figure 6 diseases-11-00179-f006:**
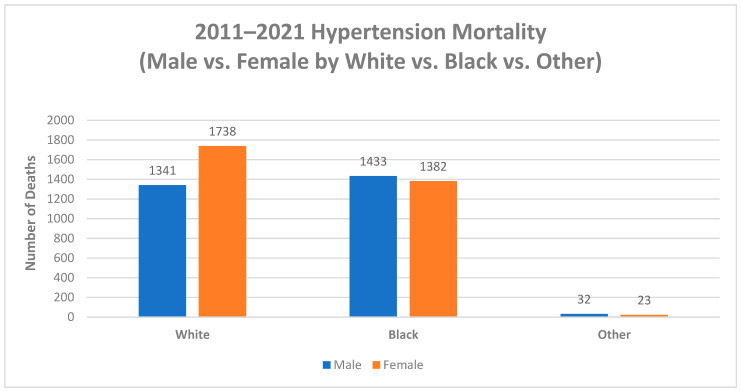
Hypertension deaths by Sex and Race, Mississippi, 2011–2021.

**Table 1 diseases-11-00179-t001:** Trends in Alzheimer’s Deaths, Mississippi, 2011–2021.

Characteristic	No. of Cases (Age-Adjusted Rates)		AAPC (95% CI)	Trend Segment 1 (95% CI)		Trend Segment 2 (95% CI)	
	2011	2021	2011–2021	Years	APC	Years	APC
**Gender**							
Male	310 (30.7)	459 (34.4)	4.61 (1.1 to 9.5)	2011–2021	4.61 (1.1 to 9.5) *		
Female	664 (34.9)	1236 (62.2)	6.78 (4.50 to 9.94)	2011–2021	6.78 (4.50 to 9.94) *		
**Race**							
Black	190 (27.3)	373 (47.1)	7.32 (4.89 to 10.97)	2011–2021	7.32 (4.89 to 10.97) *		
White	782 (35.9)	1315 (55.9)	5.90 (2.80 to 10.07)	2011–2021	5.90 (2.80 to 10.07) *	2019–2021	6.0 (0.1 to 9.6) *
Other	**** (7.8)	7 (11.6)	4.19 (−5.5 to 20.2)	2011–2014	35.0 (7.1 to 236.4) *	2014–2021	−6.8 (−27.7 to −0.9) *
**Age (yrs)**							
0–14	0 (0)	0 (0)	0 (0)	0 (0)			
15–44	0 (0)	0 (0)	0 (0)	0 (0)			
45–64	12 (0.3)	20 (0.5)	5.99 (2.2 to 11.2) **	2012–2021	5.99 (2.2 to 11.2)		
65+	962 (33.2)	1675 (182.3)	6.1 (2.7 to 10.7)	2011–2021	6.1 (2.7 to 10.7)		

Note: * Significant AAPC, or APC, Average Annual Percentage Change (AAPC); **** Variables less than 5 observations are not listed in MSTAHRS; ** AAPC began in 2012 due to too few observations to fit the specified model.

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
