# Peer review of "Mortality Trends in Alzheimer’s Disease in Mississippi, 2011–2021"

_diseases, 2023, doi:10.3390/diseases11040179_

Round 1

Reviewer 1 Report

Comments and Suggestions for Authors

This is a well-written epidemiological study examining mortality trends related to ALZ specifically in the southern state of Mississippi between 2011-2021. The data are well-presented and described and discussed appropriately. The authors spend some time discussing which states showed similar changes, noting several southern states - a map of the US showing changes by state would be helpful to visualize the changes and highlight the importance of this issue in this region, not just Mississippi. 

The authors should be cautious about statements such as the first sentence of the conclusion (lines 394-396) - the "especially in Mississippi" was not really shown in this study as the authors did not do a similar analysis of nearby/neighboring states and then show that MS was particularly sensitive - just be careful in wording as all of these states - the so-called "stroke belt" are now showing similar patterns with ALZ, and have several of the same lifestyle factors that probably contribute to the issue.

While well-written, sections 4 and 5 could be removed without any loss from the manuscript - they are fine if this is mostly a review of the topic, but the article could be more tightly focused without these sections which take away from the major contributions being made from the epidemiological data.

Author Response

Reviewer #1

Reviewer’s Comments

Response

Line(s)

1.     A map of the US showing changes by state would be helpful to visualize the changes and highlight the importance of this issue in this region, not just Mississippi. 

Corrected

100

2.     The authors should be cautious about statements such as the first sentence of the conclusion (lines 394-396) - the "especially in Mississippi" was not really shown in this study as the authors did not do a similar analysis of nearby/neighboring states and then show that MS was particularly sensitive - just be careful in wording as all of these states 

Corrected

251-252

3.     Sections 4 and 5 could be removed without any loss from the manuscript - they are fine if this is mostly a review of the topic, but the article could be more tightly focused without these sections which take away from the major contributions being made from the epidemiological data.

Corrected

Removed

Reviewer 2 Report

Comments and Suggestions for Authors

The article by Jones et al. is not a scientific work. It could be published in a popular science edition for schoolchildren (not even for students), but not in a scientific journal. In fact, the presented article is a compilation of information from popular science or social websites dedicated to Alzheimer's disease (AD). Of the 34 references, only four references (references [3-6]) are the works published in scientific journals, and the authors cite these articles in one short sentence (Numerous research findings have suggested that lifestyle factors play a major role in the onset of Alzheimer’s disease [3-6].). No self-respecting scientific journal would publish this kind of opus.

Now in more detail by sections:

Introduction.

Not all the sources cited by the authors are credible.

For example, there is this the following quote in the article: «In the United States, Alzheimer’s disease is the sixth leading cause of death and the fifth leading cause of death among people 65 and older living in the United States [7]»). Reference [7] is a web-article on the website of an organization providing assistance to people with AD. The author of this web-article is Senior Copyrighter (!), who states that “According to our research team's analysis 134,242 people died from Alzheimer's disease in 2020, the most recent year of data available and Alzheimer's is the fifth leading cause of death in the United States for people over 65, etc.”. The information about what kind of research team this is, who are the members of this team, where their analysis was published and how it was carried out is provided neither in the reference [7] nor in the article by Jones et al.

2. Epidemiology of Alzheimer’s Disease & Mortality Trends

In this section and its subsections, the authors provide statistical information from the Mississippi Statistically Automated Health Resource System and the CDC website. All the authors’ work consisted only of constructing Excel graphs based on these data. This is the level of coursework for first-year students, but not a scientific article. To explain these graphs (which highlight differences in AD incidence and mortality by race, gender, and age), the authors could have done a serious job of reviewing the available research on the topic and providing well-grounded evidence for why this or that factor affects mortality.  Instead, the authors cite some short popular science web-articles. For example, lines 135-139: «Blacks also have a higher mortality trend rate than Whites and other races in Mississippi. Explanations for these differences in the magnitude of increases in mortality rates include the prevalence of health conditions among Blacks, such as heart disease, discrimination within the healthcare systems, and issues navigating the healthcare system [21]». Reference [21] is a brief article on the informational web portal https://www.webmd.com/, written by a freelance writer (!) (albeit reviewed by a professional doctor). This webmd article does not provide a single reference to any scientific research neither.

Line 76: Figures 3 and 4 are referenced before Figures 1 and 2.

3. Etiology; 4. Diagnosis and Prognosis; 5 Stages of Alzheimer’s Disease; 6. Risk Factors; 7. Prevention

There is no need to review these sections separately. All of them are copy-paste from several informational web-sites; most of the web-articles that Jones et al. cite are even without authorship. A particularly egregious case is Section 5, which takes up almost two pages and is entirely a copy-paste of just one website dedicated to AD (reference [30]). Moreover, for some reason, the authors put a ref. [30] after each sentence. The remaining sections look similar. That is, the work of the authors on these sections consisted only of copying text from several websites into their article.

8. Conclusion.

The conclusion (like the entire article) does not contain any new information. This is just a repetition of what has already been published and has long been known.

Author Response

Reviewer #2

Reviewer’s Comments

Response

Line(s)

1.     Not all the sources cited by the authors are credible.

Corrected

282-349

2.     Line 76: Figures 3 and 4 are referenced before Figures 1 and 2.

Corrected

Line 76

3.     There is no need to review sections 4&5 separately.

Corrected

Removed

4.     The conclusion does not contain any new information. 

Corrected

264-269

Reviewer 3 Report

Comments and Suggestions for Authors

Review article on "Mortality Trends in Alzheimer’s Disease in Mississippi, 2011-2021" recommended for  minor revision. 

Following points must be addressed before submitting revised  version.

Case study carried out is pretty older, include the latest study.

How this review is differ from published articles?

Comments on the Quality of English Language

Extensive editing of English language required

Author Response

Reviewer #3

Reviewer’s Comments

Response

Line(s)

1.     Case study carried out is pretty older, include the latest study.

If referring to the trend analysis (2011-2021), the availability of the data is based on the release of data by Vital Statistics. Therefore, the trend is based on the most recent data release.

N/A

2.     How this review is differ from published articles?

Corrected

67-72

Round 2

Reviewer 2 Report

Comments and Suggestions for Authors

The authors have improved the article, it can be published.